# S2 Peptide-Conjugated SARS-CoV-2 Virus-like Particles Provide Broad Protection against SARS-CoV-2 Variants of Concern

**DOI:** 10.3390/vaccines12060676

**Published:** 2024-06-18

**Authors:** Chang-Kyu Heo, Won-Hee Lim, Ki-Beom Moon, Jihyun Yang, Sang Jick Kim, Hyun-Soon Kim, Doo-Jin Kim, Eun-Wie Cho

**Affiliations:** 1Rare Disease Research Center, Korea Research Institute of Bioscience and Biotechnology (KRIBB), 125 Gwahak-ro, Youseong-gu, Daejeon 34141, Republic of Korea; ckheo40@kribb.re.kr (C.-K.H.); whlim4028@gmail.com (W.-H.L.); 2Department of Functional Genomics, KRIBB School of Bioscience, Korea University of Science and Technology, Daejeon 34141, Republic of Korea; 3Plant Systems Engineering Research Center, Korea Research Institute of Bioscience and Biotechnology (KRIBB), Daejeon 34141, Republic of Korea; irony83@kribb.re.kr (K.-B.M.); hyuns@kribb.re.kr (H.-S.K.); 4Infectious Disease Research Center, Korea Research Institute of Bioscience and Biotechnology (KRIBB), Daejeon 34141, Republic of Korea; jhyang@kribb.re.kr; 5Synthetic Biology and Bioengineering Research Center, Korea Research Institute of Bioscience and Biotechnology (KRIBB), Daejeon 34141, Republic of Korea; sjick@kribb.re.kr; 6Chungbuk National University College of Medicine, 194-15 Osongsaengmyeong 1-ro, Osong-eup, Cheongju-si 28160, Republic of Korea; golddj@cbnu.ac.kr

**Keywords:** SARS-CoV-2, universal vaccine, virus-like particles, conserved S2, peptide conjugation, broadly neutralizing antibody

## Abstract

Approved COVID-19 vaccines primarily induce neutralizing antibodies targeting the receptor-binding domain (RBD) of the SARS-CoV-2 spike (S) protein. However, the emergence of variants of concern with RBD mutations poses challenges to vaccine efficacy. This study aimed to design a next-generation vaccine that provides broader protection against diverse coronaviruses, focusing on glycan-free S2 peptides as vaccine candidates to overcome the low immunogenicity of the S2 domain due to the N-linked glycans on the S antigen stalk, which can mask S2 antibody responses. Glycan-free S2 peptides were synthesized and attached to SARS-CoV-2 virus-like particles (VLPs) lacking the S antigen. Humoral and cellular immune responses were analyzed after the second booster immunization in BALB/c mice. Enzyme-linked immunosorbent assay revealed the reactivity of sera against SARS-CoV-2 variants, and pseudovirus neutralization assay confirmed neutralizing activities. Among the S2 peptide-conjugated VLPs, the S2.3 (N1135-K1157) and S2.5 (A1174-L1193) peptide–VLP conjugates effectively induced S2-specific serum immunoglobulins. These antisera showed high reactivity against SARS-CoV-2 variant S proteins and effectively inhibited pseudoviral infections. S2 peptide-conjugated VLPs activated SARS-CoV-2 VLP-specific T-cells. The SARS-CoV-2 vaccine incorporating conserved S2 peptides and CoV-2 VLPs shows promise as a universal vaccine capable of generating neutralizing antibodies and T-cell responses against SARS-CoV-2 variants.

## 1. Introduction

Coronavirus disease 2019 (COVID-19) is an infectious respiratory disease triggered by severe acute respiratory syndrome coronavirus 2 (SARS-CoV-2), which has had severe social and economic impacts worldwide [1]. Since the COVID-19 outbreak in December 2019, extensive efforts have been made to develop prevention and treatment strategies. Numerous SARS-CoV-2 vaccine candidates have been developed, including mRNA, viral vector-based, and subunit vaccines [2]. Many vaccines targeting the SARS-CoV-2 spike (S) protein have been developed and authorized for use [3,4,5,6,7,8].

However, the emergence of variants, such as BA.1 (Omicron), and newer variants, such as BA.2, BA.4, and BA.5, which can evade vaccine-induced immunity, is of considerable concern [9,10,11,12]. Moreover, the future trajectory of coronavirus variants remains uncertain, and some zoonotic coronaviruses have the potential to cause pandemics [13,14]. Therefore, developing a universal vaccine to address potential new or reemerging coronaviruses is necessary [15].

Several strategies for developing SARS-CoV-2 universal vaccines have focused on eliciting broadly protective antibody responses. One strategy involves presenting various mutant coronavirus spikes or receptor-binding domain (RBD) antigens on a single nanoparticle, aiming to activate cross-reactive B cells [16,17]. Coronavirus vaccines based on nanoparticles, incorporating various RBD antigens from different strains, have exhibited broad neutralizing effects against SARS-CoV-2 variants. An alternative approach involves engineering an S protein that incorporates common mutations, aiming to induce strong neutralizing responses effective against all variants of concern. A pan-vaccine antigen, referred to as Span, was developed after studying the homology across 2675 SARS-CoV-2 S protein sequences. This vaccine features residues that are frequently found at specific sites, reflecting broad cross-clade consistency in sequence evolution [18]. Administering the Span vaccine to mice induced a broad immune response, effective against multiple variants, even those that appeared post-vaccine development. Using conserved parts of the S protein is another strategy for developing universal vaccines. This strategy focuses on the fact that the S1 region, which contains the RBD, exhibits high sequence variability, and the S2 region, which is responsible for cell fusion, is highly conserved among variants. The functional properties of the S2 region and its conserved sequence make it a promising candidate for universal vaccines and fusion inhibitors [19,20].

The S2 subunit, comprising 588 amino acids (residues 686–1273), includes a fusion peptide (FP), two heptad repeats (HR1 and HR2), a transmembrane (TM) domain, and a cytoplasmic tail (CT) domain. This subunit facilitates viral fusion and entry. During viral entry, the S2 subunit undergoes highly dynamic conformational changes. Following the proteolytic cleavage of the S1 subunit, the S2 subunit embeds the fusion peptide (FP) into the membrane of the target cell. This action exposes the prehairpin coiled coil within the HR1 domain, initiating an interaction with the HR2 domain that leads to the formation of a six-helical bundle with the HR1 trimer. This process draws the viral and cellular membranes closer together, facilitating viral fusion and entry [21,22].

In the prefusion state, the spike features three flexible hinges in its stalk, enabling widespread conformational diversity within the fully glycosylated system via interactions between glycans and between glycans and lipids [23].

When glycosylated, the spike protein exhibits a significant expanded dynamic range, allowing the spike to explore a broader area of the host cell surface [23]. In contrast, in the postfusion state, the S2 surface is decorated with five N-linked glycans along the long axis, exhibiting regular spacing. Glycans present in the pre- and postfusion states may shield the S2 region from host immune responses [24,25].

Extensive studies have been conducted on SARS-CoV-2 S2-targeted vaccines, which aim to induce broadly neutralizing antibodies, taking into account the conformational properties of the S2 region [19,26]. Initially, efforts have been made to enhance S2 antigen expression using DNA vaccines [27] or virus-like particles (VLPs) displaying S2 glycoproteins [28]. However, the presence of N-linked glycans in the S2 subunit has been suggested to impede humoral immune responses [23,24,25,29]. Furthermore, mRNA vaccines expressing glycan-free S2 antigens have been developed to address the limitation of hindering the exposure of S2 epitopes [30]. These vaccines have been reported to effectively induce neutralizing antibodies against SARS-CoV-2 variants. Additionally, recombinant protein vaccines, which are designed to mimic the postfusion state structure of the S2 subunit without N-linked glycans, have been explored [31]. Although precise mapping of epitopes targeted by these S2-based vaccines has not been conducted [27,28,29,30,31], these results indicate that the conformational characteristics of the S2 proteins in the vaccine play a crucial role in eliciting neutralizing responses against the virus.

Analyzing the structural characteristics of virus-neutralizing antibody epitopes provides valuable insights into the development of potential vaccines. Analyses of SARS-CoV-2 neutralizing antibody epitopes are primarily performed in convalescent patients. Most neutralizing antibodies against the S2 subunit, including CC40.8, B6, S2P6, and CV3-25 [32,33,34,35], have been generated against the epitopes of flexible hinges in the stalk covering the peptide sequence from 1142Q to 1159H [32]. This epitope sequence, which has attracted attention as a target element for vaccine development [19,20], corresponds to the sequence between the N-glycans N1134 and N1158. hMab5.17 is another monoclonal antibody targeting the conserved region of the SARS-CoV-2 spike protein. Although derived from SARS-CoV-1-infected mice, it can neutralize SARS-CoV-2 infection and protect animals from SARS-CoV-2 challenges [36]. The epitope was mapped between S1161 and S1175 of the SARS-CoV-2 S protein, which corresponds to the peptide sequence between the N-glycans N1158 and N1173. Additionally, a novel anti-S2 neutralizing antibody was previously characterized [37], which was obtained from mice immunized with inactivated SARS-CoV-2. The epitope was identified as peptide segment F1109-V1133, located between the heptad repeat 1 (HR1) and stem-helix (SH) regions. This region corresponds to the stalk region of the pre- or postfusion S antigen located between the N-glycans N1098 and N1134.

The stalk structure of the S antigen shows six N-linked glycans (N1074, N1098, N1134, N1158, N1173, and N1194) [38]. Despite the dramatic conformational changes in the S2 subunit during the virus–cell fusion process, the peptide sequences between these N-glycans are highly conserved in SARS-CoV-2 variants, and these regions are exposed to the outer side of the prefusion state or the 6-helical bundle structure of the postfusion state of the S2 subunit (Appendix A) [24,39]. Moreover, the sequence analysis of these five peptides showed that some can form stable alpha-helix structures with high antigenicity (Appendix A).

Based on these findings, we postulated that antibodies elicited against the peptide sequences between these glycans could act as neutralizing antibodies. Therefore, in this study, five peptides were synthesized corresponding to the regions between the six glycans of the stalk structure of the S antigen and presented as antigens to assess their potential to induce neutralizing antibodies. SARS-CoV-2 VLPs composed of NP, E, and M proteins without the S antigen were used as carriers for peptide conjugation to maintain the immunogenicity of these short peptide antigens. Mice were vaccinated with the S2 peptide–VLP vaccine via subcutaneous injection into the footpad. The binding properties of immunized serum antibodies to variant spike antigens were analyzed. Furthermore, the neutralizing efficacy against SARS-CoV-2 was evaluated by pseudoviral assays using total serum. The antibodies elicited by S2 peptides were characterized, and whether the S2 peptide could be used as a component of a universal SARS-CoV-2 vaccine was investigated.

## 2. Materials and Methods

### 2.1. Design and Synthesis of Antigenic Peptides for the SARS-CoV-2 S2 Region

Peptide sequences located between six N-glycans (N1074, N1098, N1134, N1158, N1173, and N1194) surrounding the C-terminus of the SARS-CoV-2 S protein (Wuhan-Hu-1 strain, NCBI Accession Number: YP_009724390.1) were used as antigenic peptide candidates. The designed antigenic peptides were named S2.1 (S:1083–1097), S2.2 (S:1102–1125), S2.3 (S:1135–1157), S2.4 (S:1159–1172), and S2.5 (S:1174–1193). Modified peptides with an additional cysteine residue at the C-terminus of the selected peptide sequences were synthesized for conjugation to the carrier protein, either VLP or bovine serum albumin (BSA) (Dandicure, Ochang, Chungcheongbuk, Republic of Korea). For fine epitope mapping of antibodies or antisera reactive to the S2.3 peptide, S2.3pN (S:1135–1148) and S2.3pC (S:1144–1157) were synthesized with cysteine modification at the C-terminus and then conjugated to BSA.

### 2.2. Production of SARS-CoV-2-NEM VLPs

SARS-CoV2-NEM VLPs assembled with three SARS-CoV-2 viral structural proteins (M, E, and NP) were produced in plants based on the method described by Moon et al. [40]. Briefly, VLPs were transiently expressed through agroinfiltration of pBYR2fp-MIRESE and pBYR2fp-NPFLAG vectors that expressed the viral genes M, E (MIRESE), and NPflag (NFLAG) in *N. benthamiana* leaves. The leaves were harvested and mechanically ground in phosphate-buffered saline (PBS). The insoluble material was precipitated by centrifugation at 10,000× *g* for 20 min at 4 °C. The soluble extracts, which included self-assembled VLPs consisting of NP, M, and E proteins, were then concentrated five-fold using an ultrafiltration unit with a 5 kDa molecular weight cutoff (Sigma-Aldrich, Burlington, MA, USA). Subsequently, VLPs were purified using sucrose gradient ultracentrifugation. The self-assembly of VLPs was confirmed using Western blotting and transmission electron microscopy (TEM).

### 2.3. Conjugation of S2 Peptides to CoV-2-NEM VLPs

The antigenic peptide derived from the SARS-CoV-2 S2 region was attached to CoV-2-NEM VLPs using the heterobifunctional crosslinker, sulfosuccinimidyl 4-(N-maleimidomethyl)cyclohexane-1-carboxylate (Sulfo-SMCC; Thermo Fisher Scientific, Waltham, MA, USA, Cat# A39268), following the guidelines provided by the manufacturer. VLPs were pretreated with Tris(2-carboxyethyl)phosphine (TCEP; Sigma-Aldrich, Cat# 646547) and alkylated with iodoacetamide (Sigma-Aldrich, Cat# I1149) before the conjugation reaction to block free thiol groups on the VLP surface. Conversely, cysteine-containing synthetic peptides were dissolved in conjugation buffer (PBS containing 5 mM EDTA) and reduced using an immobilized TCEP disulfide reduction gel to maximize conjugation efficiency. An amount of 40 µL of crosslinker Sulfo-SMCC (5 mg/mL) in deionized double distilled water was mixed with 1 mL of VLP (1 mg/mL) in the conjugation buffer. The mixture was allowed to react for 30 min at room temperature. The excess crosslinker was removed using a desalting column (5 mL Zeba spin column, Thermo Fisher Scientific, Cat# 89892) equilibrated with conjugation buffer. Then, 320 μg of VLP-SMCC (0.6 mg/mL) was mixed with 220 μg of cysteine-containing S2 peptide (10 mg/mL) and allowed to react for 30 min at room temperature. Finally, the peptide–VLP conjugates were concentrated, and the buffer was changed to PBS using an ultrafiltration device (MWCO 10K Merck Millipore, Burlington, MA, USA, Cat# UFC901024). The S2 peptides were conjugated to BSA using the same conjugation method to analyze their antibody binding properties.

### 2.4. Characterization of S2 Peptide-Displaying VLPs (or BSA)

S2 peptide-conjugated VLPs (or BSA) were analyzed by 10% sodium dodecyl sulfate-polyacrylamide gel electrophoresis (SDS-PAGE) to verify the efficiency of conjugation. To further confirm this, Western blotting and enzyme-linked immunosorbent assay (ELISA) were conducted using anti-S2 antibodies, as detailed subsequently. Whether or not the VLPs retained their structure after SMCC-mediated conjugation to the S2 peptides was confirmed using TEM analysis (Eulji University School of Medicine, Seongnam, Gyeonggi, Republic of Korea). Briefly, a 10 µL solution of the VLPs or VLP peptides was dropped onto carbon-coated, glow-discharged copper grids for 2 min. Then, a 10 µL solution of 2% uranyl acetate was dropped onto the grids for negative VLP staining, followed by drying to adsorb the VLPs. The grids were then analyzed using a TEM H-7600 (Hitachi, Tokyo, Japan) at a magnification of 200,000× *g*. Images of each VLP were collected.

### 2.5. Mouse Immunization

The Institutional Animal Care and Use Committee of the Korea Research Institute of Bioscience and Biotechnology (KRIBB) reviewed and approved the animal care and experimental protocols (approval number: KRIBB-AEC-22204), and experiments were conducted in accordance with the Guidelines for Animal Experiments of the KRIBB. Five-week-old female BALB/c mice were obtained from Orient Bio (Seongnam, Gyeonggi, Republic of Korea) and housed in a pathogen-free facility. After 1 week of acclimation, blood samples were collected from the mice’s tails before immunogen injection. All groups of mice (*n* = 3 or 6) were vaccinated three times (priming, first, and second boost) at 2-week intervals. For immunization, 50 μL of the antigen–adjuvant mixture was injected into the footpads of mice. The mixture consisted of 10 μg of S2 peptide-conjugated VLP or VLP in PBS (25 μL) and an equal volume of TiterMax Gold adjuvant (Sigma-Aldrich, Cat# T2684). Additionally, SARS-CoV-2 S2 (Sino Biological, Cat# 40590-V08B, Beijing, China), S_trimer_ (Sino Biological, Cat# 40589-V08H2B), or inactivated SARS-CoV-2 was immunized using the same protocol. Inactivated SARS-CoV-2 was prepared by propagating the SARS-CoV-2 virus in Vero cells and treating it with beta-propiolactone for inactivation [41]. Blood samples were collected from the tail vein of the mice two weeks after each injection. Serum was separated and preserved at −80 °C until analysis.

### 2.6. Western Blot Analysis

S2 peptide-conjugated VLPs or BSA proteins were separated on a 10% SDS-PAGE gel and subsequently transferred to a polyvinylidene difluoride (PVDF) membrane. This membrane was blocked using PBST (PBS with Tween 20) with 5% skim milk, then incubated with primary antibodies in blocking buffer for 2 h at room temperature. Following this, the membrane was washed three times with PBST, then incubated with horseradish peroxidase (HRP)-conjugated goat anti-mouse IgG antibody (1:10,000; Cell Signaling Technology, Danvers, MA, USA) diluted in PBST with 5% skim milk. The protein bands were then detected using enhanced chemiluminescence reagents. The primary antibodies used were SARS-CoV-2 S2 monoclonal antibody (clone 17F706R; Novus Biologicals, Cat# NBP2-90999, Centennial, CO, USA) and SARS-CoV-2 nucleocapsid monoclonal antibody (clone 1G6) [41].

### 2.7. ELISA

Each antigen (100 ng) was diluted in 100 µL of PBS and coated onto MaxiSorp 96-well plates (Thermo Fisher Scientific, Cat# 44240) in duplicate, followed by incubation at 4 °C overnight. The plate was blocked with 5% skim milk in Tris-buffered saline containing 0.1% Tween 20 and incubated for 2 h at room temperature. Primary antibodies were diluted in blocking solution and added to each well (100 μL per well), followed by incubation for 90 min at 37 °C. Then, an HRP-conjugated goat anti-mouse IgG antibody (Cell Signaling Technology) diluted to 1:2500 in blocking solution containing 1% BSA was added to each well (100 μL/well). The plate was incubated at 37 °C for 90 min. The reaction was developed using 1-Step™ Ultra TMB-ELISA Substrate Solution (Thermo Fisher Scientific, Cat# 34028). After incubation for 10 min, the reaction was stopped using 0.16 M H_2_SO_4_, and the reactivity was determined by measuring the optical density at 450 nm (OD_450_) using a Molecular Device microplate reader VersaMax (Molecular Devices, San Jose, CA, USA). Peptide-conjugated BSA or VLP and viral antigens were used as coating antigens. The viral antigens were purchased from Sino Biological, including SARS-CoV Spike ECD (Cat# 40634-V08B), SARS-CoV-2 (2019-nCoV) Spike ECD (D614G) (Cat# 40589-V08B4), MERS-CoV Spike ECD (Cat# 40069-V08B), HCoV-OC43 Spike ECD (Cat# 40607-V08B), HCoV-HKU1 Spike ECD (Cat# 40606-V08B), SARS-CoV-2 Alpha B.1.1.7 Spike ECD (Cat# 40589-V08B6), SARS-CoV-2 Gamma P.1 Spike ECD (Cat# 48589-V08B10), SARS-CoV-2 Delta B.1.617.2 Spike ECD (Cat# 40589-V8B16), SARS-CoV-2 Omicron B1.1.529 Spike ECD (Cat# 40589-V08B33), SARS-CoV-2 nucleocapsid protein (NP; Cat# 40588-V08B), and SARS-CoV-2 S2 ECD (S686-P1213; Cat# 40590-V08B). Immunized sera were diluted to 1:10,000 or serially diluted in a blocking buffer containing 1% BSA to be used as primary antibodies. Commercial antibodies were diluted according to the ratios indicated in each figure, including NBP2 (anti-SARS-CoV-2 S2 monoclonal antibody, Novus Biologicals, Cat. NBP2-90999, Immunogen SARS-CoV-2 S (1142-1158)) and MA5 (anti-SARS-CoV-2 S2 monoclonal antibody clone 1A9, Thermo Fisher, Cat. MA5-35946, Immunogen: SARS-CoV-S Delta10 (1029-1192)). To produce S2P6 antibody [34] in the form of human IgG1, the sequences for the variable regions of both the heavy and light chains (Integrated DNA Technologies, Coralville, IA, USA) were synthetized. These sequences were then inserted into the pCEP-WPRE-hCH (hIgG1) vector for the heavy chain and the pCEP-WPRE-hCK vector for the light chain using the NEBuilder HiFi DNA Assembly system (New England Biolabs, Ipswich, MA, USA), following the protocols previously outlined [42]. After constructing the vectors for both chains, they were co-transfected into ExpiCHO cells (Thermo Fisher Scientific, Waltham, MA, USA). The recombinant IgG molecules were subsequently produced and purified according to the referenced protocol [42].

### 2.8. Splenocyte T-Cell Activation Assay

The frequencies of IFNγ-producing T-cells in splenocytes from immunized mice was evaluated using mouse IFNγ ELISpot kits (BD Biosciences, San Jose, CA, USA). Briefly, splenocytes were isolated from the spleens of immunized or naïve mice. The isolated splenocytes were suspended in RPMI 1640 medium and plated at a density of 1 × 10^6^ cells/well onto ELISpot plates precoated with anti-IFNγ antibody. Then, they were stimulated with VLPs at 37 °C in a CO_2_ incubator. After 2 days, the cell culture supernatants were collected, and the plates were washed. The detection antibody was diluted in PBS with 10% fetal bovine serum, added to each well (100 µL per well), and incubated for 2 h at room temperature. After washing, streptavidin-HRP was added to each well. After incubation for 1 h at room temperature, the plate was washed and AEC substrate (BD Biosciences) was added. The reaction was stopped by adding deionized water. The plate was air-dried. Finally, the spots were measured using an ELISpot plate reader (Cellular Technology Ltd., Cleveland, OH, USA). The IL-4 assay was performed in activated splenocytes using a mouse IL-4 ELISA kit (BD Biosciences). Briefly, an IL-4 standard solution or cell culture supernatant from stimulated splenocytes was added to plates coated with anti-IL4 antibody and incubated for 2 h at room temperature. After washing, the detection antibody and streptavidin-HRP reagent were added and incubated for 1 h at room temperature. The plate was then washed, and 1-Step™ Ultra TMB (100 µL per well) was added and incubated for 30 min. The reaction was stopped using a stop solution (0.16 M H_2_SO_4_), and OD_450_ was measured. The IL-4 concentration in splenocyte culture supernatants was calculated using a standard curve.

### 2.9. SARS-CoV-2 Pseudovirus-Based Neutralization Assay

Pseudoviruses carrying the spike protein of SARS-CoV-2 variants were generated using pLV-Spike (prototype, Wuhan-Hu-1), pLV-SpikeV1 (D614G variant), and pLV-SpikeV12 (Omicron BA.2) vectors (InvivoGen, San Diego, CA, USA). Lenti-X 293T cells (Takara Bio, San Jose, CA, USA) were cultured in nonphenol red Dulbecco’s Modified Eagle’s Medium (DMEM) (Thermo Fisher Scientific) supplemented with 10% heat-inactivated fetal bovine serum and co-transfected with the pLV-Spike plasmids, pCDH-EF1a-eFFly-eGFP (Addgene plasmid #104834), and psPAX2 (Addgene plasmid #12260) using TransIT-VirusGEN transfection reagent (Mirus, Madison, WI, USA). psPAX2, which encodes the necessary virion packaging proteins, was generously provided by Didier Trono. Additionally, pCDH-EF1a-eFFly-eGFP, which encodes reporter proteins, was kindly provided by Irmela Jeremias. Pseudoviruses were harvested from the cell culture supernatants by filtration 3 days after transfection. Then, they were divided into single-use aliquots and stored at −80 °C until use. The viral titer of the SARS-CoV-2 pseudovirus was determined using a Lenti-X™ p24 Rapid Titer Kit (Takara Bio). The degree of viral infection was confirmed. Pseudoviruses with an adjusted relative light unit of 10^4^ or greater were mixed with phenol red-free DMEM and diluted to 1:50 with VLP peptide-immunized sera to prepare a total mixture of 100 μL. The mixture was incubated for 1 h at 37 °C. HEK-Blue-hACE2-TMPRSS2 cells (InvivoGen, San Diego, CA, USA) were added to a plate at a density of 3 × 10^4^ cells and treated with the virus–antibody mixture. The cells were then incubated for 24 h at 37 °C in a CO_2_ incubator. After incubation, cells were lysed using the One-Glo^®^ Luciferase Assay System (Promega, Madison, WI, USA). The level of infection inhibition was measured by detecting luciferase activity.

### 2.10. Competitive ELISA

A competitive ELISA was performed using HRP-conjugated S2P6 antibodies against mouse sera immunized with S2.3-VLPs to analyze the activity of S2P6-like broadly neutralizing antibodies, which react with the SH region of the Spike S2 region [34]. A MaxiSorp 96-well plate was coated with 20 ng of SARS-CoV-2 S2 protein in PBS and incubated overnight at 4 °C. A mixture of 10 ng HRP-conjugated S2P6 monoclonal antibody and competitor sera was prepared in PBS to a final volume of 100 μL. The mixture was then added to a plate and incubated for 2 h at 37 °C. After washing, the reaction was developed using 1-Step™ Ultra TMB-ELISA Substrate Solution. The OD_450_ of the reaction was measured. S2P6 monoclonal antibody was used as a competitor. The competition rate was assessed to determine the standard reaction for competition. HRP-conjugated S2P6 was prepared using an HRP conjugation kit (Abcam, Cambridge, UK, Cat# ab102890).

### 2.11. Statistical Analysis

All graphs, calculations, and statistical analyses were performed using GraphPad Prism 10, version 10.0.0 (GraphPad Software, San Diego, CA, USA). All values are presented as mean ± standard deviation (SD).

## 3. Results

### 3.1. Selection of Antigenic Peptide Sequence for a Universal Vaccine against SARS-CoV-2

The stalk domain of the S protein contained six N-glycan modification sites (N1074, N1098, N1134, N1158, N1173, and N1194) (Figure 1A and Appendix A). Short glycan-free S2 peptides were selected, corresponding to the sequences between five N-linked glycan modifications surrounding the stalk region, as antigenic epitopes to develop an S2-based vaccine capable of generating potent antibodies that can disrupt the protein refolding process from the prefusion to postfusion state and impede viral entry. These peptides were successively designated as S2.1, S2.2, S2.3, S2.4, and S2.5 (Figure 1A and Appendix A). Sequences spanning from N1074 to N1194 exhibited a high level of conservation across most SARS-CoV-2 variants, except for specific mutations observed in the Alpha B.1.1.7 variant (D1118H) or the Gamma (P1) variant (V1176F) [43].

The presentation of short S2 peptides on carrier proteins or nanoparticles is essential for using them as immunogens [44]. Various strategies, such as genetic fusion, tag coupling, and chemical conjugation, can be used for this purpose [45]. In this study, we opted for the chemical conjugation approach because of its capacity to facilitate multiple displays of the peptide on carrier proteins, thereby enhancing the immunogenicity of the peptide antigen [46]. SMCC, a heterobifunctional amine-to-sulfhydryl crosslinking reagent, was selected as the chemical crosslinker. The presence of primary amines on the surface of protein carriers or nanoparticles allowed them to react with amine-reactive crosslinkers, enabling the coupling of peptides. The antigenic peptides were designed to include a cysteine at the C-terminus for conjugation with the crosslinker. However, two of the candidate S2 peptide sequences already contained cysteine residues (C1082 and C1126). If these residues were included in the synthetic peptides with an additional cysteine at the C-terminus, the two cysteines within the candidate peptide could participate in the conjugation reaction, resulting in multiple conformations of the carrier-peptide conjugation. The sequences of S2.1 and S2.2 peptides were adjusted to exclude internal cysteines to avoid such complexity. The S2.1 peptide sequence was trimmed to exclude residue C1082 from the antigenic peptide, resulting in S (1083-1097). The S2.2 peptide sequence was selected as S (1102-1126), and C1126 was used as the C-terminal cysteine. The final antigenic peptide sequences were determined (Table 1).

The suitability of the candidate peptides as antigenic epitopes was evaluated by analyzing their physicochemical properties and predicted structures (Table 1 and Appendix A). Based on the findings of Hopp and Wood [47], the average hydrophilicity values of the S2.3 and S2.5 peptides were 0.47 and 0.32, respectively (Table 1). In comparison, the hydrophilicity of the S2.1, S2.2, and S2.4 peptides was relatively lower, ranging from −0.46 to 0.19. Additionally, the ratio of hydrophilic residues to the total number of amino acids was approximately 50% for the S2.3 and S2.5 peptides, whereas it was relatively low (approximately 30%) for the S2.1, S2.2, and S2.4 peptides. The predicted structures of the antigenic candidate peptides generated using Alphafold2 revealed that S2.3 and S2.5 exhibited helical structures (Appendix A). In contrast, S2.1, S2.2, and S2.4 adopted loop-shaped structures with some helical elements retained.

The selected epitope sequences were synthesized with over 99% purity. Peptide-conjugated BSA was prepared and examined using SDS-PAGE, Western blotting, and ELISA to evaluate the suitability of the peptide conjugation reaction for displaying epitope peptides on carriers. Except for the S2.2 peptide, the S2 peptides exhibited good solubility in the conjugation reaction buffer. Dimethyl sulfoxide (DMSO) was added to the reaction buffer at a final concentration of 10% to enhance the solubility of the S2.2 peptide, which exhibited low hydrophilicity (Table 1). After the conjugation reactions, the reaction mixtures were analyzed using 10% SDS-PAGE under reducing conditions and Coomassie blue staining (Figure 1B and Appendix A). The SMCC-bound BSA exhibited a shift from the position of BSA, indicating significant crosslinker binding. After the conjugation reaction with the peptides, additional shifts in the protein bands were observed, confirming an increase in the molecular weight due to the attachment of multiple peptides to BSA. However, in the case of the S2.2 peptide conjugation reaction, no shift in the protein bands was observed, indicating that peptide conjugation did not occur. This lack of conjugation may be attributed to the low hydrophilicity and subsequent low solubility of the S2.2 peptide in the conjugation reaction solution, resulting in a low yield of conjugation. Furthermore, the antibody responses of the conjugates were examined. Monoclonal antibodies (NBP2, MA5, and S2P6), known to interact with the SH domain of the S protein [34], which is a part of the S2.3 peptide, exhibited reactivity with the S2.3 peptide–BSA conjugate in Western blotting and ELISA assays, indicating the successful conjugation of antigenic peptides to BSA (Figure 1B (middle) and C).

Additionally, the reactivity of S2 peptide–BSA conjugates was analyzed using antisera against inactivated SARS-CoV-2 virus (IAV) or S antigens. The antisera generated against IAV displayed high titers of anti-NP and S antibodies (Appendix A). These sera showed substantial reactivity toward the S2 antigen in ELISA (Figure 1D left) and significant reactivity with S2.3 peptide–BSA conjugates. Furthermore, the antisera generated against the S2 antigen or S_trimer_, which showed high reactivity toward the S2 antigen, exhibited significant reactivity with S2.3 peptide–BSA conjugates (Figure 1D, middle, right). Additionally, these sera significantly reacted to S2.5 peptide–BSA. These results provide evidence that conjugation of the S2 peptide with the carrier protein using the SMCC crosslinker effectively presents antigenic epitopes that can react with anti-S2 antibodies elicited by exposure to the whole antigen or virus.

### 3.2. Vaccination with S2 Peptide-Conjugated CoV-2-VLPs Elicits Antigen-Specific Humoral and Cellular Immune Responses in BALB/c Mice

To develop a universal vaccine, the selected peptides were conjugated from the S2 region to SARS-CoV-2 VLPs, which are composed of NP, E, and M proteins but lack the S antigen produced in *N. benthamiana* plants [40]. The verification of these VLPs confirmed their similar shape and size to native SARS-CoV-2 virions, despite the lack of spike structures on their surface [40]. SARS-CoV-2 VLPs offer unique advantages as a vaccine platform compared with other nanoparticles, such as ferritin, M2, and other viral VLPs [45]. They possess nanostructures that closely resemble natural SARS-CoV-2, effectively stimulating cellular and humoral immune responses. Additionally, they are free of viral nucleic acids, ensuring excellent safety and thermal stability, and they can be produced and stored on a large scale. Furthermore, their surfaces can be modified to provide additional functionalities.

The S2 antigenic peptides containing C-terminal-free cysteine (Table 1 and Figure 2A) were conjugated to VLPs using the bifunctional crosslinking agent SMCC. The conjugated VLPs were analyzed using 10% SDS-PAGE, followed by Coomassie staining and Western blotting to confirm the successful conjugation of the peptides to VLPs (Figure 2B and Appendix A). VLPs appeared as prominent bands of high-molecular-weight polymerized proteins on the Coomassie blue-stained gels. However, protein bands corresponding to NP, E, and M were not visible. However, Western blot analysis using anti-NP antibodies revealed the presence of NP-related proteins in VLPs, including monomers (~48 kDa), truncated forms (~28 kDa), and high-molecular-weight polymers. The intensity of the bands corresponding to monomeric NP proteins was reduced upon conjugation with SMCC. This may be due to antigenic epitope masking caused by chemical modification. The conjugation of S2 peptides to VLPs resulted in a shift of protein bands toward higher-molecular-weight bands, as observed in immunostaining with anti-S2 antibody NBP2, which recognizes the S2.3 peptide. Even after the peptide conjugation reaction, the VLP retained its characteristic particle features, as confirmed by the TEM analysis (Figure 2C). The characteristics of the S2 peptide-displaying VLPs were also analyzed using ELISA (Figure 2D). Anti-NP antibodies showed a high level of reactivity with the VLPs. However, anti-NP antibody responses were reduced by conjugation of the SMCC and S2 peptides, as confirmed by Western blotting (Figure 2D left). The reaction of the anti-S2 antibody, NBP2, confirmed the presence of S2 peptides on VLPs at a high density (Figure 2D right).

The immunogenicity of S2 peptide-conjugated VLPs was evaluated by vaccination in BALB/c mice following the schedule outlined in Figure 2E. Each group of three mice was immunized with either VLPs or S2 peptide-conjugated VLPs formulated with an adjuvant at priming, primary boost, and secondary boost. The VLP immunogens were administered via subcutaneous injection into the footpad at a dose of 10 μg per injection. Mouse sera were collected before priming and after the secondary boost.

The humoral immune responses against immunogens were analyzed by measuring the total IgG levels against the immunogens using ELISA (Figure 2F). Unmodified VLPs elicited little IgG responses. However, strong antibody responses were observed against the peptide–VLP conjugates, including S2.1-VLP, S2.3-VLP, S2.4-VLP, and S2.5-VLP. The humoral immune response against S2.2-VLPs was very low, similar to the response against VLPs. This may be due to the low efficiency of S2.2 peptide conjugation. The cellular immunity induced by vaccination with S2 peptide–VLP conjugates was confirmed using the IFNγ ELISpot assay (Figure 2G). Splenocytes from mice immunized with S2.3-VLP or naïve unimmunized mice were stimulated with VLP. The production of IFNγ was assessed using the ELISpot assay. Splenocytes from S2.3-VLP-immunized mice showed an increase in IFNγ production in response to increasing VLP concentrations, indicating activation of the Th1 response. Splenocytes from naïve mice did not exhibit any response. Furthermore, splenocytes from S2.3-VLP-immunized mice produced IL-4 by the stimulation with VLP (Figure 2H); however, the increase in IL-4 was low, indicating that activation of the Th2 response was not a major immune response to S2.3-VLP immunization. These findings confirm that SARS-CoV-2 VLPs conjugated with S2 peptides effectively serve as antigens that induce SARS-CoV-2 S2-specific humoral and cellular immunity.

ELISA was performed against VLPs and peptide-conjugated BSA to confirm the specificity of antibodies elicited by S2 peptide-conjugated VLPs (Figure 3A). The sera obtained from immunization with S2 peptide–VLP conjugates, except for S2.2-VLP, exhibited minimal reactivity toward unmodified VLPs and strong reactivity toward the corresponding S2 peptide–BSA conjugates. No reactivity to BSA was observed, indicating the specific reactivity of the antisera to each conjugated S2 peptide. The sera from mice immunized with S2.2-VLP displayed little reaction to S2.2-BSA, consistent with the abovementioned results showing that the antiserum against S2.2-VLP did not elicit specific antibodies. One of the antisera against S2.2-VLP showed some reactivity against VLP, although at a very low level.

The reactivity of antisera against the SARS-CoV-2 S2 antigen was evaluated. The S2 antigen expressed in HEK293 cells was used as a coating antigen to mimic the viral antigenic structure, as it retains posttranslational modifications, such as N-glycan modifications and disulfide bonding. As verified through their reactivity to peptide-conjugated BSA, the S2 peptide-conjugated VLPs (excluding S2.2-VLPs) effectively induced the production of S2 peptide-specific antibodies. However, responses to the S2 antigen were observed only in antisera against the S2.3 or S2.5 peptide VLPs (Figure 3A). Additional mouse immunization was performed using S2.3- and S2.5-VLPs (*n* = 3 for each immunization) to validate these results. The antibody characteristics were confirmed using ELISA, yielding consistent results (Appendix A). Responses to NPs were detectable after VLP immunization. However, little response to antisera against S2 peptide–VLP conjugates was observed (Figure 3A).

These results indicate that the SARS-CoV-2 VLP, devoid of spikes and featuring a smooth surface, is inefficient in inducing humoral immune responses against itself. However, when the VLP is coupled with antigenic peptides, leading to the display of peptide epitopes on the surface, it can effectively elicit strong antibody responses. Moreover, maintaining a secondary structure similar to that maintained by viral protein antigen peptides, such as S2.3 and S2.5, can induce antibody responses that recognize the original antigen.

The reactivity of S2.3- and S2.5-VLP immune sera against the S2 antigen was further investigated. The reactivity to the S2 antigen in the serially diluted sera of S2.3- or S.2.5-VLP was analyzed, revealing consistently high reactivity to the S2 antigen (Figure 3B), with similar endpoint titer values observed (Figure 3C). Interestingly, the antibody response in S2.3-VLP sera exhibited variability among individual mice, whereas immunization with S2.5-VLP led to a more consistent level of antibody response (Figure 3B).

### 3.3. S2 Peptide-Conjugated VLPs Elicit Broadly Reactive Antibodies against SARS-CoV-2 Variants

The reactivity of S2 peptide–VLP immune sera against the S antigens of various SARS-CoV-2 variants, including D614G, Alpha, Gamma, Delta, and Omicron BA.1, was further analyzed to evaluate the potential of S2.3- or S2.5-VLP as a universal vaccine against SARS-CoV-2 variants. The reactivity against S antigens of other betacoronaviruses (β-CoVs), including SARS-CoV and MERS-CoV, and endemic human coronaviruses (HCoV-HKU1 and HCoV-OC43) was examined for comparison. Antisera immunized with S2.3-VLPs showed high reactivity against S antigens of SARS-CoV-2 variants, encompassing different strains, such as D614G, Alpha, Gamma, Delta, and Omicron BA.1, as well as SARS-CoV. However, these antisera did not exhibit reactivity against the S antigens of MERS-CoV, HCoV-HKU1, and HCoV-OC43 (Figure 4A left). These reactivity characteristics largely reflect differences in epitope sequences. The S2.3 peptide sequence is 100% identical in all SARS-CoV-2 variants and SARS-CoV, forming an SH structure (Figure 4B and Appendix A). Consequently, antibodies generated by immunization with the S2.3 peptide are expected to react equally with all S protein structures containing an identical sequence. In contrast, the sequences of MERS-CoV and HCoV variants corresponding to the S2.3 peptide exhibited significant differences, with less than 20% sequence identity to the S2.3 region of SARS-CoV-2 variants or SARS-CoV (Figure 4B). The sequences of the S2.5 region are almost identical in SARS-CoV-2 variants and SARS-CoV. Therefore, sera from immunization with S2.5-VLPs are expected to react to SARS-CoV and SARS-CoV-2 variants. Sera from immunization with S2.5-VLPs displayed high reactivity to the S antigens of SARS-CoV-2 variants, including D614G, Alpha, and Delta (Figure 4A right). However, the reactivity against S antigens of Gamma and Omicron was relatively low. The reactivity to the S antigen of SARS-CoV was relatively low. Antisera against S2.5-VLPs showed no reactivity with the S antigens of MERS-CoV, HCoV-HKU1, and HCoV-OC43 (Figure 4A right).

The efficacy of the S2 peptide–VLP antisera was further evaluated for neutralizing activity against SARS-CoV-2 variants (Wuhan, D614G, and Omicron BA.2) using pseudovirus neutralization assays (Figure 4C). Sera from S2.3- or S2.5-VLP-immunized mice demonstrated an average of 50% neutralizing activity against the SARS-CoV-2 variants. However, the virus neutralization capacity of S2.3- or S2.5-VLP-immunized sera differed from their S antigen-binding capacity. The S2.3-VLP-immunized sera showed no significant difference in binding to SARS-CoV-2 variant S antigens. However, its neutralizing capacity against the Omicron variant was slightly lower than that of the Wuhan and D614G variants. In contrast, S2.5-VLP-immunized sera showed low reactivity to the Omicron S antigen. However, their neutralizing capacity against the Omicron variant was similar to that of the Wuhan and D614G variants.

VLP-immunized sera, which did not contain antibodies targeting viral S antigens, did not exhibit viral neutralizing capacity (Figure 4C). Similarly, sera from S2.1- or S2.4-VLP-immunized mice, which lack antibodies against viral S2 antigen (Figure 3A), showed no neutralization activity against the SARS-CoV-2 variants (Figure 4C).

### 3.4. C-Terminal Exposure of S2.3 Peptide Confers an Epitope That Induces S2P6-like Broadly Neutralizing Antibodies against SARS-CoV-2

The S2.3 peptide sequence includes the segment of S (1148–1156), which has been identified as an epitope of numerous neutralizing antibodies against SARS-CoV-2, including S2P6 [34]. Therefore, the S2.3 peptide was expected to elicit neutralizing antibodies with similar activity to that of S2P6. Competitive ELISA for S antigen was conducted using HRP-labeled S2P6 to assess whether S2.3-VLP immune sera include broadly neutralizing antibodies, such as S2P6 (Figure 5A). Furthermore, a standard curve for competitive inhibition ELISA by the S2P6 antibody was established (Appendix A), and the S2P6-like antibody response of S2.3-VLP immune sera was estimated. An amount of 20 μL of S2.3-VLP sera exhibited activity at a level of approximately 0.2 μg/mL of S2P6 antibody (Figure 5B and Appendix A), which closely approached the S2P6-like activity of an equal amount of S2 immune sera (Figure 5B and Appendix A).

However, considering that the response of S2.3-VLP sera to S2.3-BSA was somewhat higher than the response to the S2 antigen (Figure 3A), it is expected that S2.3-VLP may present two or more different epitopes, including the S2P6 epitope. Two different sequences of the S2.3 peptide (S2.3pN, S2.3pC) were synthesized to test this hypothesis (Table 1 and Figure 5C), and their responses to immune sera or S2-specific antibodies were examined. S2.3pC, which includes the S (1148-1156) sequence, exhibited high reactivity to S2P6, NBP2, and MA5 antibodies known to react with the S2 subunit (Figure 5D). However, S2.3pN, which lacks the S (1149-1156) sequence, showed no reactivity to these antibodies. In contrast, S2.3-VLP and S2 immune sera demonstrated reactivity to peptides, S2.3pN, and S2.3pC (Figure 5D). Notably, S2.3-VLP immune sera exhibited slightly higher reactivity to the S2.3pN peptide compared with S2.3pC (Figure 5D), whereas S2 immune sera exhibited higher reactivity to the S2.3pC peptide (Figure 5D). S2.3-VLP immune sera exhibited higher endpoint titers to the S2.3pN peptide than S2.3pC, and their reactivities to S2.3pC varied between individual mouse sera. However, the responses to S2.3pN were similarly highly induced (Figure 5E,F).

These results prompted us to reevaluate the conjugation reaction of antigenic peptides to VLPs. The peptide–VLPs used in this study were synthesized by adding cysteine to the C-terminus of the peptide for conjugation with VLPs. Therefore, in S2.3-VLP antigens prepared through conjugation reaction, the S (1148–1156) sequence was positioned close to the VLP, with relatively limited exposure compared with the N-terminal side of the S2.3 sequence. Whether the induction of antibodies to the S (1148–1156) sequence could be increased by positioning cysteine for conjugation on the N-terminus was assessed by immunizing with VLPs conjugated with S2.3 analogs, including cysteine at the C- or N-terminus, namely, S2.3.1 and S2.3.2 (Figure 5G), and examining the level of induction of S2P6-like antibodies. Sera immunized with S2.3.1-VLP or VLP-S2.3.2 induced antibodies at similar levels for S2.3pN and S2.3pC peptides (Figure 5H and Appendix A). However, S2P6-like activity was induced at much higher levels in VLP-S2.3.2-immunized sera in which the N-terminus of the S2.3 peptide was employed to the conjugation reaction, whereas its C-terminus was free (Figure 5I and Appendix A). Furthermore, VLP-S2.3.2-immunized sera showed higher reactivity to S antigen variants than S2.3.1-VLP-immunized sera (Figure 5J).

### 3.5. Immune Sera Elicited with Peptide Lacking N-Terminal Sequence Containing V1176 of S2.5 Peptide Showed Enhanced Reactivity toward S Antigen Variants

The low reactivity of the S2.5-VLP antiserum to the Gamma variant S antigen may be due to the influence of the V1176 to F mutation located at the N-terminus of the S2.5 peptide (Figure 4B). For a detailed analysis, another S2.5 sequence-related peptide, S2.5.2, starting from V1177 without the V1176F mutation of the Gamma variant, was employed, and the antigenic specificity of immune sera was characterized (Figure 6A). Amino acid residues from N1194 to Q1201 were added at the C-terminus of the S2.5 peptide to compensate for the shortened length of S2.5.2 and add spacer amino acids between VLP and S2.5 peptide (Table 1). Additionally, the S2.5.1 peptide, a peptide of an extended form of S2.5, was synthesized and used for the reactivity of antisera against S2.5-immunized sera. Peptides were conjugated using cysteine at the C-terminus. The S2.5-VLP antisera exhibited a high antibody response to both antigens (Figure 6B,C). Although responses to the S2.5.1 peptide were consistently high in all S2.5 immune sera, responses to the S2.5.2 peptide varied significantly between individuals. Interestingly, in sera immunized with S2.5.2-VLP, the response to the S2.5.2 peptide significantly increased, surpassing the reactivity observed for the S2.5.1 peptide (Figure 6D). The endpoint titer against S2.5.2-BSA was increased in S2.5.2-VLP antisera (Figure 6E), and the reactivity against S variants was increased in S2.5.2-VLP antisera compared with that in S2.5-VLP antisera. However, the reactivity against SARS S antigen was decreased (Figure 6F and Figure 4A, right).

### 3.6. Neutralizing Capacities of the S2 Peptide–VLP Antisera Were Induced to the Level of the Neutralizing Activity of the S2 Antigen Antisera

The neutralizing activities of VLP-S2.3.2 and S2.5.2-VLP antisera against SARS-CoV-2 variants (Wuhan, D614G, Omicron BA.2) were further evaluated (Figure 7A). The antisera generated against VLP-S2.3.2 showed 50% inhibition of infection against all three virus variants, whereas the antisera against S2.3.1-VLP showed a lower level of neutralization activity, reflecting their binding activity to S antigens (Figure 5J). Sera immunized with S2.5.2-VLP also exhibited an average neutralizing activity of 50% against SARS-CoV-2 variants (Figure 7A).

The neutralizing activities of these S2 peptide–VLP-immunized sera were similar to those of S2P6, a well-known broadly neutralizing anti-SARS-CoV-2 S2 antibody [34], and S2 antigen-immunized sera. The neutralizing activity of S2P6 was assessed in a dose-dependent manner (Appendix A). S2P6 effectively neutralized the three SARS-CoV-2 variants, consistent with previous reports [34], which showed that, at concentrations of 10–20 μg/mL, S2P6 was able to neutralize 50%–60% of infections with these variants, like the neutralizing activity of VLP-S2.3.2 and S2.5.2-VLP antisera. The neutralizing activity of S2 antigen-immunized sera prepared according to the same immunization protocol as S2 peptide–VLPs was in the range of 50%–80%, which was slightly higher than that of S2 peptide–VLP-immunized sera (Figure 7B). Unsurprisingly, sera from mice immunized with S_trimer_ containing multiple epitopes in the S1 or S2 domains showed up to 100% neutralization activity against SARS-CoV-2 variant pseudoviruses (Appendix A).

## 4. Discussion

The emergence of SARS-CoV-2 and its variants was not sudden. They exhibit morphological and genetic similarities with other members of the β-CoV family, such as SARS, MERS, and some seasonal HCoVs. The high rate of RNA virus evolution plays a significant role in transforming these β-CoVs into potentially dangerous pathogens. Furthermore, their ability to cause infections in a wide range of animals and humans contributes to their evolutionary process. Therefore, the concerns about the future emergence of these viruses and the potential development of new threats emphasize the importance of developing a universal vaccine capable of targeting β-CoVs, including SARS-CoV-2.

This study aimed to design a universal vaccine tailored to combat SARS-CoV-2 variants. The vaccine formulation was designed to include conserved epitopes capable of inducing humoral and cellular immune responses to SARS-CoV-2 variants. Conserved peptide sequences from the S2 subunit of SARS-CoV-2 were selected to induce broadly neutralizing antibody production. Additionally, VLPs consisting of NP, E, and M proteins with conserved sequences across SARS-CoV-2 variants were employed to introduce T-cell epitopes essential for cellular immunity. Furthermore, incorporating VLPs enabled us to include their properties as immune cell-stimulating nanoparticles, which have been proposed as crucial components of vaccines [19].

The conformation of the S2 subunit of the SARS-CoV-2 virus undergoes dynamic changes during the process of viral entry into cells and is structurally constrained by glycans, which can pose challenges for antibody induction. The reported antibodies targeting the S2 domain are limited to the FP and SH sites, corresponding to the most exposed N-terminus or hinge structure of the S2 antigen [20]. Therefore, previous studies have not focused much on antibody responses to the S2 subunit and have primarily emphasized its role as a cellular immune epitope [20,28]. However, in our study, we hypothesized that although the stalk domain of the S protein is not suitable for the induction of specific antibodies, preexisting antibodies targeting epitope sequences included in the stalk domain could access the space between N-glycans during the conformational changes of virus–cell fusion. As a result, these antibodies could bind to specific epitopes in the stalk domain, potentially interfering with viral cell fusion. Based on this hypothesis, a vaccine was designed to generate antibodies that target epitopes located between N-linked glycans in the stalk region. Among the five peptides studied, S2.3 (^1135^NTVYDPLQPELDSFKEELDKYF K^1157^) and S2.5 (^1174^ASVVNIQKEIDRLNEVAKNL^1193^), which exhibited high hydrophilicity and adopted a helical structure, successfully induced antibodies capable of recognizing viral S antigens. Moreover, these antibodies exhibited neutralizing activity, effectively inhibiting the infection of SARS-CoV-2 variants.

In fact, it was anticipated that the S2.3 peptide sequence, which includes epitopes for well-known broadly neutralizing antibodies that bind to the SH domain, could elicit neutralizing antibodies. However, identifying the broadly neutralizing potential of antisera against the S2.5 peptide sequence is a novel finding of this study. The S2.5 sequence encompasses the HR2 region, which plays a crucial role in viral cell fusion [20]. No reports of neutralizing antibodies specifically targeting the S2.5 sequence have been reported in studies involving convalescent patients or individuals who have received the SARS-CoV-2 vaccine. Only peptides targeting this region with fusion inhibitory properties have been developed, suggesting a potential strategy for inhibiting viral infection [20]. This study highlights the importance of inducing antibodies against the S2.5 sequence to enhance the effectiveness of the vaccine. It is necessary to generate monoclonal antibodies targeting S2.5 peptides and evaluate their efficacy in inhibiting viral infection to further validate these findings. This additional investigation can provide conclusive evidence regarding the effectiveness of the S2.5 peptide-conjugated SARS-CoV-2 VLP vaccine as a potential inhibitor of viral infection.

A notable advantage of this study is the use of VLPs produced through a plant expression system. Plant systems have gained attention as cost-effective methods for large-scale production of eukaryotic proteins, making them an attractive platform for next-generation vaccine development [48]. In this study, SARS-CoV-2 VLPs produced using a plant expression system were effectively employed as vaccine components that elicit cellular immunogenicity, although they do not possess S protein spikes. Furthermore, the vaccine efficacy of CoV-2-VLPs has been enhanced by the conjugation of potent antigenic peptides, resulting in the induction of broadly neutralizing antibodies against SARS-CoV-2 variants. A cost-effective vaccine-producing platform consisting of a plant expression system and short synthetic peptides can be another hope for combating infectious diseases.

When designing a peptide–VLP conjugate vaccine, it is important to ensure that B-cell epitopes on the surface of VLPs are appropriately displayed to induce antibody responses. As observed in the immune response to S2.3- or S2.5-derived peptide-conjugated VLPs, it is crucial to position epitope peptides on the VLPs away from structural constraints to facilitate the induction of antibodies against epitopes. It is necessary to add long spacers between epitopes and VLPs or expose the termini of sequences crucial for antibody response toward the solution, ensuring that antibody access is not structurally restricted.

Additional research is needed to effectively harness the advantages of peptide–VLP conjugate vaccines for clinical applications. Although the reactivity and neutralizing capacity of antibodies generated through immunization with S2.3- and S2.5-VLPs were confirmed, it is crucial to perform in vitro and in vivo experiments to validate the efficacy of the vaccine. Infection experiments involving passive immunization using immunized sera or direct immunization using disease models, such as hamsters or ACE2-expressing mice, in Biosafety Level 3 (BSL-3) laboratories are necessary. Furthermore, further research is needed to optimize vaccine components. Translating the necessary assemblies for vaccine construction into clinically applicable forms and evaluating their effectiveness are essential.

This study is of significant importance as it provides a straightforward approach to developing a universal vaccine targeting viruses undergoing complex mutations. Further research will enable the development of a vaccine suitable for clinical application.

## 5. Conclusions

This study showed that the cryptic S2 epitopes, S2.3 (^1135^NTVYDPLQPELDS FKEELDKYFK^1157^) and S2.5 (^1174^ASVVNIQKEIDRLNEVAKNL^1193^), which are masked by N-glycan in the S protein of SARS-CoV-2, can be used as antigenic peptides to elicit broadly neutralizing antibodies. Additionally, VLPs assembled with SARS-CoV-2 structural proteins M, E, and NP not only stimulate immune cells due to their unique particle characteristics but also activate cellular immunity by presenting SARS-CoV-2 T-cell epitopes. Therefore, developing a SARS-CoV-2 vaccine incorporating conserved S2 peptides and CoV-2-VLPs holds promise as a universal vaccine strategy capable of inducing neutralizing antibodies and T-cell responses against SARS-CoV-2 variants.

## Figures and Tables

**Figure 1 vaccines-12-00676-f001:**
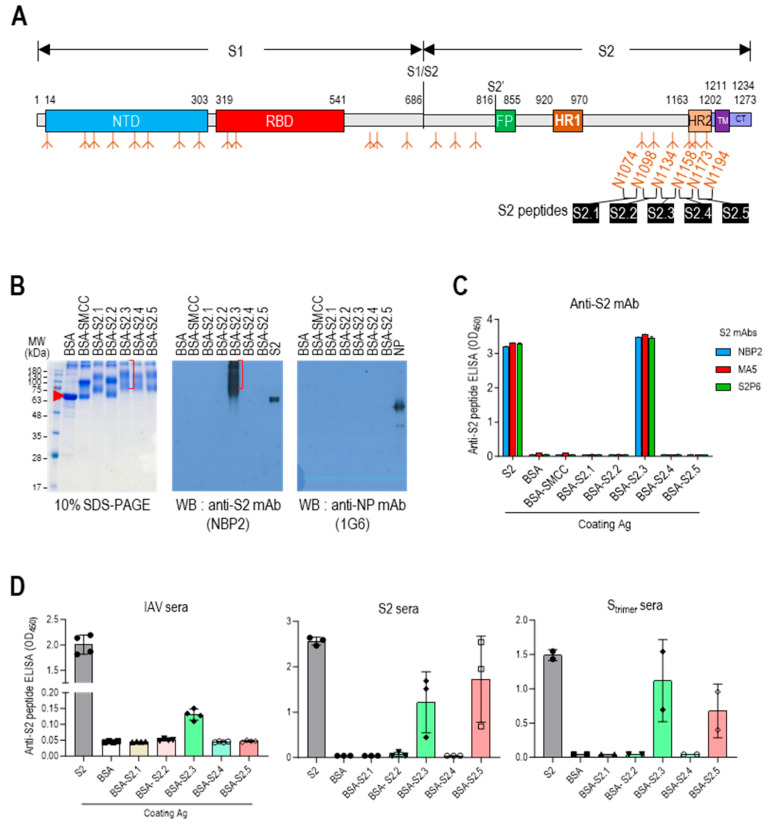
Designing antigenic peptide sequences for a broad-spectrum SARS-CoV-2 vaccine. (**A**) Diagram of the SARS-CoV-2 spike protein showing its structural components. The spike is organized into two primary domains, S1 and S2; these includes the N-terminal domain (NTD), receptor binding domain (RBD), fusion peptide (FP), heptad repeat 1 (HR1), heptad repeat 2 (HR2), transmembrane (TM), and cytoplasmic tail (CT). The specific segment boundaries of these regions are indicated, and locations of glycosylation identified through experimental methods are noted below each segment. The positions of the glycan-free S2 peptide sequences are shown. (**B**) SDS-PAGE and Western blot analysis of proteins following conjugation of S2 peptides to BSA. To confirm the formation of S2 peptide–BSA conjugates, they were probed with an anti-S2 antibody, NBP2. Anti-NP antibody probing was used as a control to assess nonspecific binding. (**C**) ELISA analysis of S2 peptide-conjugated BSA. Recombinant S2 antigen was used as a positive control. Antibodies were added at 100 ng/well. (**D**) Reactivity of antisera against inactivated SARS-CoV-2 (IAV), S2, or S_trimer_ to S2 peptide-conjugated BSA. The antisera were diluted to 1:10,000. Each dot represents the antibody response of individual mice and displays the mean ± standard deviation (SD) of replicate wells.

**Figure 2 vaccines-12-00676-f002:**
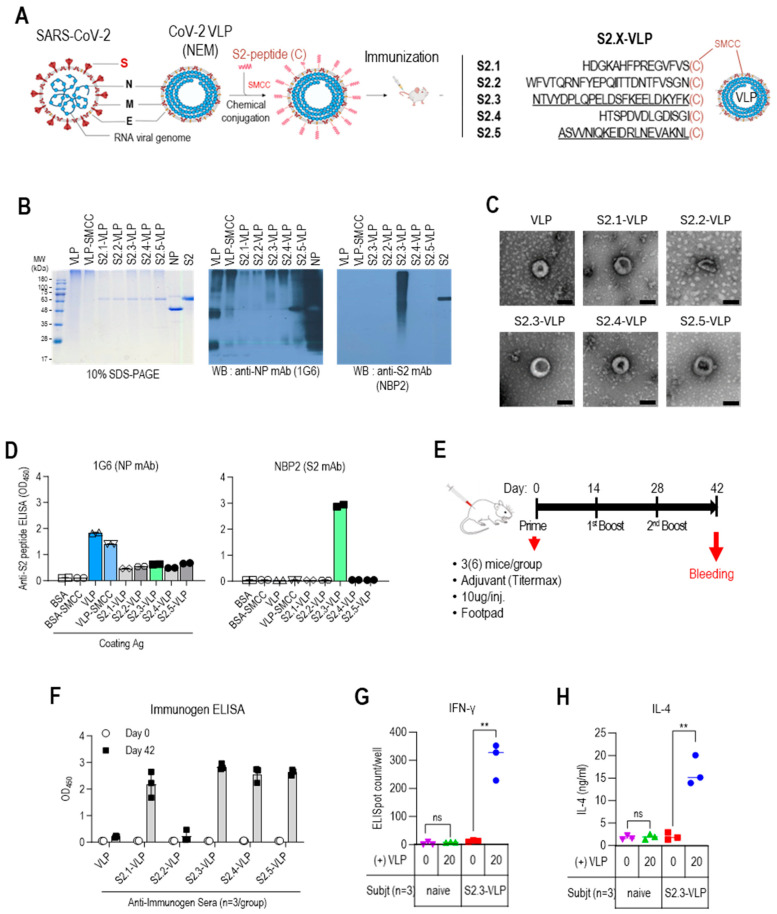
Vaccination with S2 peptide-conjugated VLPs elicits robust humoral immune responses in BALB/c mice. (**A**) S2 peptide-conjugated VLP design and S2 peptide sequences. (**B**) SDS-PAGE and Western blot analysis of S2 peptide-conjugated VLPs. The VLPs were probed with an anti-NP monoclonal antibody (1G6) to detect the presence of VLPs. The conjugated S2 peptides on the VLPs were probed with an anti-S2 monoclonal antibody (NBP2) to confirm successful conjugation. (**C**) Transmission electron micrographs (TEM) of unmodified VLPs and S2 peptide-conjugated VLPs. Representative images are shown, with the scale bar representing 100 nm. (**D**) ELISA analysis of S2 peptide-conjugated VLPs. VLPs, VLP-SMCC, S2 peptide-conjugated VLPs, and BSA (negative control) were plated and probed with SARS-CoV-2 NP-specific antibody (1G6) or S2-specific antibody (NBP2). Antibodies were added at 100 ng/well. (**E**) BALB/c immunization regimen. Groups of 3 BALB/c mice were given three immunizations with S2 peptide–VLPs or control VLPs with an adjuvant. The VLP immunogens were administered via subcutaneous injection into the footpad at a dose of 10 μg per injection. Mice were immunized at weeks 0 (prime), 2 (first boost), and 4 (second boost). Serum collection was performed before prime injection and two weeks after the second boost (day 42). (**F**) Specific total IgG responses to VLPs or S2 peptide-conjugated VLPs (S2.X-VLP). ELISA against immunogens was performed using sera (1:10,000 dilution) collected before priming and two weeks after the second boost. Each dot represents the antibody response of individual mice to VLP or S2-VLP immunization and displays the mean ± SD of replicate wells. (**G**) IFNγ ELISpot assay for splenocytes from VLP-S2.3 immunized mice. BALB/c mice (*n* = 3 per group) were immunized with S2.3-VLP or naïve. Two weeks after the second boost (day 42), pooled splenocytes were harvested and added to anti-IFNγ antibody-precoated ELISpot plates with SARS-CoV-2-NEM VLPs in duplicate. After splenocyte stimulation for 48 h, anti-IFNγ responses were developed, and spots were counted using an ELISpot plate reader. (**H**) IL4 assay for splenocytes from VLP-S2.3 immunized and naïve mice. After stimulating the splenocytes with SARS-CoV-2-NEM VLPs, the cell culture supernatant was collected and added to plates precoated with anti-IL4 antibody. The anti-IL4 response was measured, and the IL4 concentration in the splenocyte culture medium was determined using a standard curve. Each dot represents the measurement of an individual mouse, and the graph displays the mean ± SD of replicate wells. ns: *p* > 0.05, **: *p* ≤ 0.01.

**Figure 3 vaccines-12-00676-f003:**
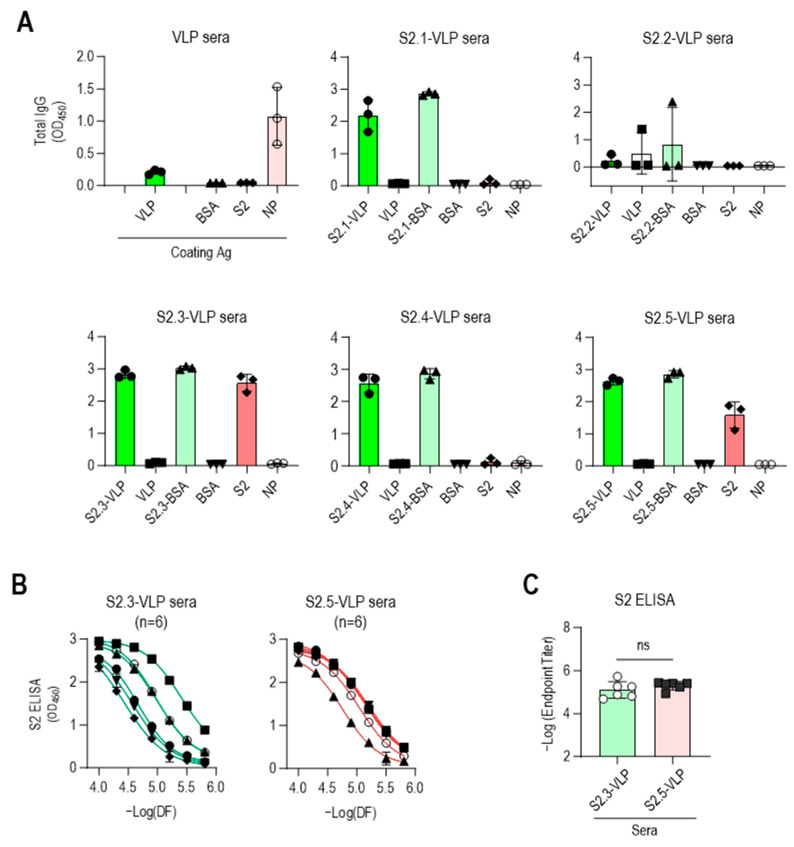
Vaccination with S2.3 or S2.5 peptide-conjugated VLPs induces SARS-CoV-2 S2 antigen-specific antibodies. (**A**) The specific immune responses of antisera against S2 peptides or SARS-CoV-2 S2 antigen were analyzed. Antisera collected after the second boost were diluted to 1:10,000 and used for analysis. Each dot represents the antibody response of an individual mouse. (**B**) Antisera obtained from immunization with S2.3-VLP or S2.5-VLP were analyzed for their specific immune responses to the S2 antigen. Antisera collected after the second boost were serially diluted and used for analysis. Serial dilutions were performed in two-fold dilutions starting at 1:10,000. Each curve represents the antibody response of an individual mouse (*n* = 6) to the S2 antigen, and each dot on the curve displays the mean ± SD of replicate wells. (**C**) SARS-CoV-2 S2 total IgG endpoint titers in serum obtained from mice vaccinated with S2.3 or S2.5 peptide-conjugated VLPs. ns: *p* > 0.05.

**Figure 4 vaccines-12-00676-f004:**
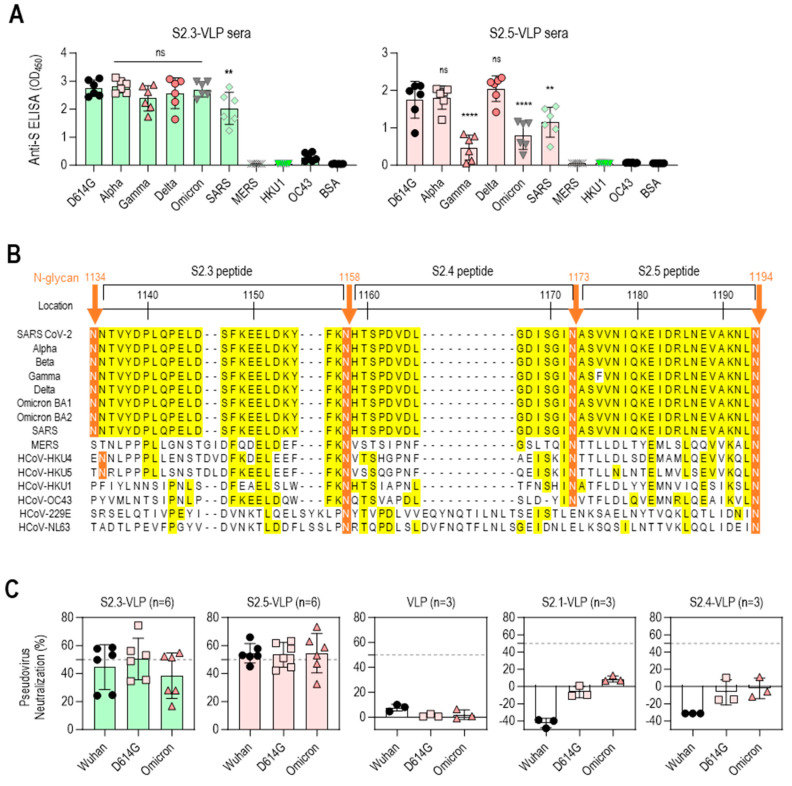
Vaccination with S2.3 or S2.5 peptide-conjugated VLPs induces broadly neutralizing antibodies against SARS-CoV-2 variants. (**A**) The total IgG response of sera from mice immunized with S2.3-VLP (left plot) or S2.5-VLP (right plot) against the spike proteins of SARS-CoV-2 variants (D614G, Alpha, Gamma, Delta, Omicron) as well as other betacoronaviruses (SARS, MERS, HKU1, and OC43) was evaluated. The immunized sera were diluted 1:10,000. ns: *p* > 0.05, **: *p* ≤ 0.01, ****: *p* ≤ 0.0001. (**B**) Amino acid alignment of the S2 peptide sequences of SARS-CoV-2 variants and related β-coronaviruses. The positions of the N-glycans are shown, and consensus sequences are highlighted in yellow. (**C**) The neutralization potential of sera obtained from mice immunized with S2.3-VLP (*n* = 6) and S2.5-VLP (*n* = 6) was assessed against the pseudotyped SARS-CoV-2 prototype (Wuhan) or variants (D614G and Omicron BA4.2). The neutralization assay was performed using sera collected on day 42 after immunization, with triplicate measurements for each sample. The pseudovirus infection without sera was used as the positive control. The unmodified VLP-immunized sera and sera from mice immunized with S2.1-VLP or S2.4-VLP were analyzed.

**Figure 5 vaccines-12-00676-f005:**
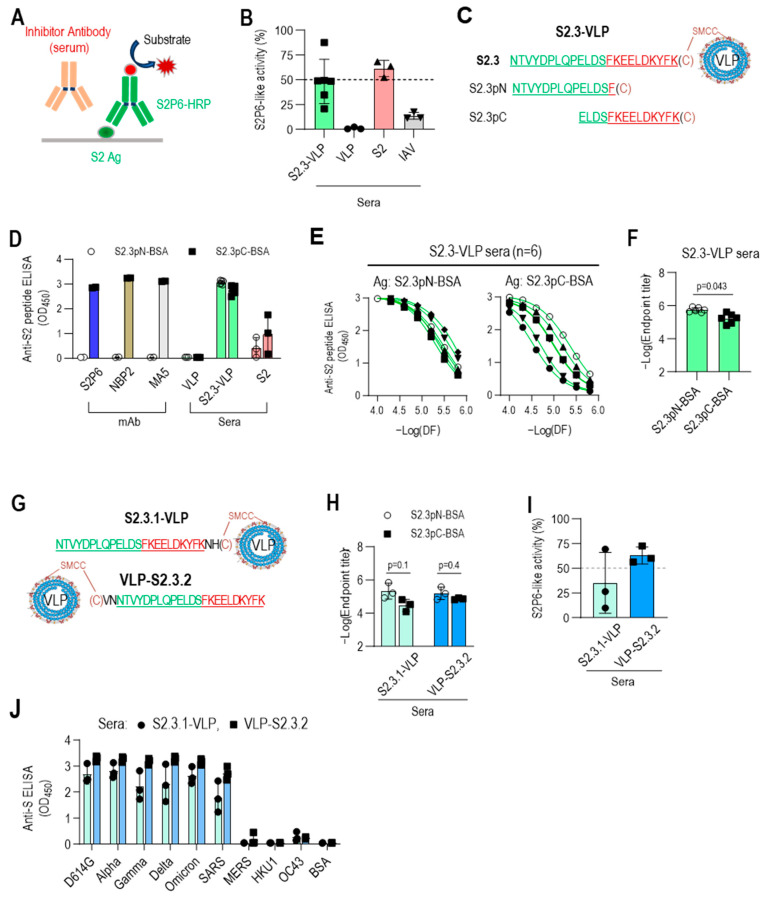
C-terminal exposure of S2.3 peptide confers an epitope that induces S2P6-like broadly neutralizing antibodies against SARS-CoV-2. (**A**) Competitive ELISA for quantifying S2P6-like activity of test sera. The recombinant S2 antigen was coated on a MaxiSorp plate. The test solution containing the sera or antibody of interest was added to the well along with HRP-conjugated S2P6. After incubation, S2P6-HRP activity was measured to determine the competitive binding between the test antibodies and HRP-conjugated S2P6 for the S2 antigen. (**B**) S2P6-like activity of VLP-S2.3, S2, or inactivated SARS-CoV-2 (IAV) immunized sera. Each dot represents the measurement of an individual mouse (*n* = 6 or 3) and displays the mean ± standard deviation (SD) of replicate wells. (**C**–**F**) Fine mapping of antigenic epitopes of S2.3-reactive antibodies or VLP-S2.3 antisera using short peptides derived from the S2.3 peptide. Two short peptides (S2.3pN and S2.3pC) were synthesized (**C**), and their conjugates to BSA served as coating antigens for detecting epitope-specific antibodies. The reactivity of S2.3 reactive antibodies, including S2P6, NBP2, and MA5 monoclonal antibodies and antisera against VLP-S2.3 or S2 antigen, was measured against these short peptides. Antibodies were added at 100 ng/well. Antisera were treated at a 1:10,000 dilution (**D**) or two-fold serial dilutions starting at 1:10,000 (**E**). Each dot represents the measurement of an individual mouse (*n* = 6 or 3) and displays the mean ± standard deviation (SD) of replicate wells. S2.3 peptide total IgG endpoint titers were also analyzed (**F**). (**G**–**J**) C-terminal exposure of S2.3 peptide on VLP confers an epitope that induces S2P6-like broadly neutralizing antibodies against SARS-CoV-2. For the exposure of the C-terminus of the S2.3 peptide on peptide–VLP conjugates, two amino acid residues were added after the S2.3 peptide as a spacer (S2.3.1-VLP) or cysteine was added at the N-terminus of the S2.3 peptide and conjugated with SMCC-linked VLPs (VLP-S2.3.2) (**G**). These peptide-conjugated VLPs were immunized following the immunization regimen (*n* = 3/each immunogen), as shown in Figure 2E, and the reactivity of antisera after the second boost against S2.3pN or S2.3pC epitopes was analyzed (**H**). Their S2P6-like activities were analyzed using sandwich ELISA, as shown in Figure 5A (**I**). Additionally, their reactivity against the S antigen of SARS-CoV-2 S variants or other β-corona viruses were examined (**J**). The antisera were diluted at 1:10,000.

**Figure 6 vaccines-12-00676-f006:**
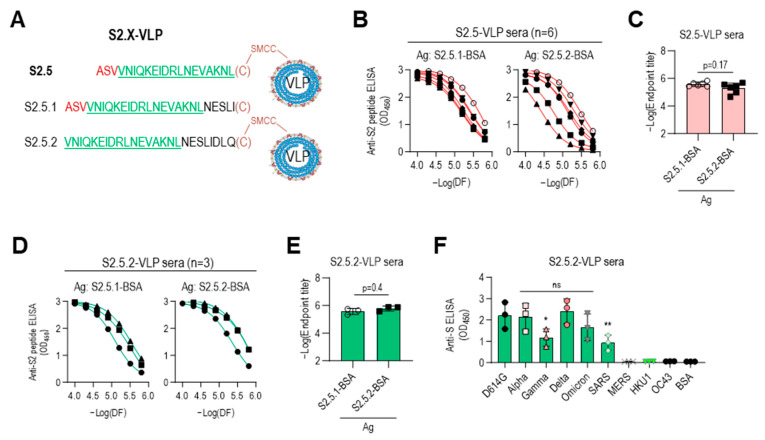
Immune sera elicited with peptides lacking an N-terminal sequence containing V1176 of the S2.5 peptide showed enhanced reactivity toward S antigen variants. (**A**–**C**) The S2.5.1 peptide, an extended S2.5 peptide from S (1174–1193) to S (1174–1198), and the S2.5.2 peptide, an S2.5 peptide with a deletion of V1176 at the N-terminus, were synthesized and conjugated to VLP or BSA (**A**), and their reactivity to S2.5-VLP-immunized sera was analyzed by endpoint titer determination. (**B**,**C**) Antisera were treated with two-fold serial dilutions starting at 1:10,000. (**D**–**E**) The reactivity of S2.5.2-VLP-immunized sera against S2.5.1 or S2.5.2 peptide was analyzed by endpoint titer determination. (**F**) Reactivity of S2.5.2-VLP-immunized sera against the SARS-CoV-2 S variants. Serum was diluted to 1:10,000. Statistical analysis was performed using one-way ANOVA, followed by Dunnett’s multiple comparison test. ns: no significance; * *p* = 0.042; ** *p* = 0.0097.

**Figure 7 vaccines-12-00676-f007:**
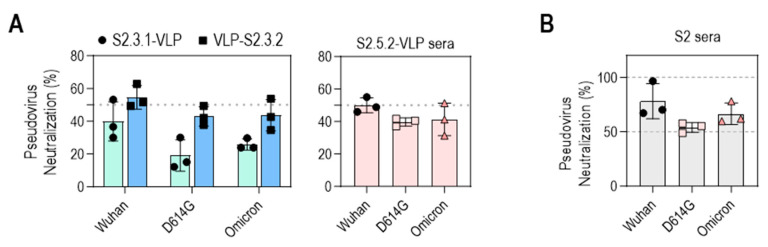
Neutralization potential of VLPs conjugated with peptides containing S2.3 or S2.5 epitopes against pseudotyped SARS-CoV-2. (**A**) Neutralization potential of S2.3 epitope (S2.3.1, S2.3.2, or S2.5.2)-conjugated VLPs. (**B**) Neutralization potential of S2 subunit-immunized sera. Pseudoviruses with SARS-CoV-2 spike proteins (Wuhan-Hu-1, D614G, or Omicron BA.2) and sera collected on day 14 after the second boost were used for neutralization assays.

**Table 1 vaccines-12-00676-t001:** Physicochemical properties of the S2 peptides used in this study.

Synthetic Peptide	Location *	Sequence	Number of Residues	Molecular Weight (g/mol)	pI ^†^	Avg. Hydrophilicity ^‡^	Hydrophilic Residues (%) ^§^
S2.1	S (1083-1097)	HDGKAHFPREGVFVS (C) **	15	1682.86	7.98	0.19	33
S2.2	S (1102-1125)	WFVTQRNFYEPQIITTDNTFVSGN (C)	24	2878.15	4.09	−0.46	38
S2.3	S (1135-1157)	NTVYDPLQPELDSFKEELDKYFK (C)	23	2819.12	4.08	0.47	52
S2.4	S (1159-1172)	HTSPDVDLGDISGI (C)	14	1425.52	3.53	0.13	36
S2.5	S (1174-1193)	ASVVNIQKEIDRLNEVAKNL (C)	20	2253.58	7.06	0.32	55
S2.3pN	S (1135-1148)	NTVYDPLQPELDSF (C)	14	1637.76	2.85	−0.04	43
S2.3pC	S (1144-1157)	ELDSFKEELDKYFK (C)	14	1790.99	4.36	0.96	64
S2.3.1	S (1135-1159)	NTVYDPLQPELDSFKEELDKYFKNH (C)	25	3070.36	4.45	0.42	52
S2.3.2	S (1133-1157)	(C) VNNTVYDPLQPELDSFKEELDKYFK ^#^	25	3032.35	4.08	0.38	52
S2.5.1	S (1174-1198)	ASVVNIQKEIDRLNEVAKNLNESLI (C)	25	2810.20	4.74	0.25	56
S2.5.2	S (1177-1201)	VNIQKEIDRLNEVAKNLNESLIDLQ (C)	25	2909.29	4.36	0.38	60

* Location: S protein sequence: YP_009724390.1. ** For the conjugation reaction, a cysteine residue was appended to the C-terminus of each S2 peptide. ^†^ The isoelectric point was estimated using the Peptide Property Calculator provided by Innovagen, a company specializing in peptide synthesis, on their website (https://www.pepcalc.com/ accessed on 14 May 2024). ^‡^ The average hydrophilicity was calculated using the Peptide Property Calculator, based on data from the study by Hopp and Woods [47]. **^§^** Hydrophilic residues (%), the proportion of hydrophilic residues relative to the total number of amino acids, was calculated using the Peptide Property Calculator. ^#^ The peptide sequences that overlap with the S2.3 and S2.5 peptides have been indicated by underlines.

## Data Availability

The datasets used and/or analyzed in this study are available from the corresponding author on reasonable request.

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
