# Peer review of "S2 Peptide-Conjugated SARS-CoV-2 Virus-like Particles Provide Broad Protection against SARS-CoV-2 Variants of Concern"

_vaccines, 2024, doi:10.3390/vaccines12060676_

Round 1
Reviewer 1 Report
Comments and Suggestions for Authors
The authors have submitted the manuscript titled "S2 Peptide-Conjugated SARS-CoV-2 Virus-Like Particles Provide Broad Protection against SARS-CoV-2 Variants of Concern".
Overall it is a well-designed study that warrants publication. I only have a few comments that need clarification.
1. The very first concern that I have is the number of animals per group in the study. The authors mention they used 3 or 6 animals per group. It is not clear which groups had 3 and which had 6 animals. I feel 3 animals per group is very little to make any statistical conclusions. In figure 3, the authors show the data from only 3 animals per group, whereas in Figure 4, there are 6 animals per group. However, the groups in both figures are the same. Then why aren't the authors showing the data from all 6 animals in figure 3 as well?
2. For the inactivated virus immunization, was the WT variant used? Was the inactivation with b-propiolactone confirmed by further propagation in the Vero cells before immunization?
3. What was the starting dilution of the sera used in ELISA? and what was the fold-dilution in the serial dilutions?
Author Response
Question1.
The very first concern that I have is the number of animals per group in the study. The authors mention they used 3 or 6 animals per group. It is not clear which groups had 3 and which had 6 animals. I feel 3 animals per group is very little to make any statistical conclusions. In figure 3, the authors show the data from only 3 animals per group, whereas in Figure 4, there are 6 animals per group. However, the groups in both figures are the same. Then why aren't the authors showing the data from all 6 animals in figure 3 as well?
--> Responses
We first immunized three animals per group with S2 peptide-conjugated VLPs (S2.1-VLP, S2.2-VLP, S2.3-VLP, S2.4-VLP, and S2.5-VLP) to determine whether they induced antibodies specific to S2 antigen (Figure 2F, Figure 3A). Among them, we found that S2.3-VLP and S2.5-VLP effectively induced S2 antigen-specific antibodies; to confirm this, we further immunized S2.3-VLP and S2.5-VLP (3 animals per group: Supplementary Figure S3). Consequently, the S2.3-VLP and S2.5-VLP immunization experiments describe the results of two immunizations of the same antigen in three mice, and in Figure 3B, Figure 4A and Figure 4C we characterized sera from six S2.3-VLP or S2.5-VLP-immunized mice.
The additional vaccinations are described in the "Results" section under 3.2.Vaccination ~ on lines 597-599 in the revised manuscript as follows: “Additional mouse immunization was performed using S2.3- and S2.5-VLPs (n = 3 for each immunization) to validate these results. The antibody characteristics were confirmed using ELISA, yielding consistent results (Figure S3).”
Question 2.
For the inactivated virus immunization, was the WT variant used? Was the inactivation with b-propiolactone confirmed by further propagation in the Vero cells before immunization?
--> Responses
The inactivated virus used for immunization was NMC-nCoV02, a strain of SARS-CoV-2 (S clade, WT variant) that was isolated from a confirmed COVID-19 patient in South Korea in February 2020. The preparation of the inactivated virus involved the following steps: The propagated viruses were inactivated by treating them with 0.5% beta-propiolactone (BPL) for 16 hours. Then the inactivated viruses were inoculated into Vero cells and incubated at 37°C in a 5% CO2 incubator. Viral inactivation was confirmed 72 hours post-infection using a plaque assay.
The process of immunizing with inactivated SARS-CoV-2 has been described in a previous study [ref 41], and sera from immunized subjects in that study were utilized in this manuscript. The description of the sera from mice immunized with inactivated viruses is correspond to lines 269-271 of the revised manuscript, in the "Materials and Methods" section under 2.5. Mouse Immunization as follows: “Inactivated SARS-CoV-2 was prepared by propagating the SARS-CoV-2 virus in Vero cells and treating it with beta-propiolactone for inactivation [41].”
Question 3.
What was the starting dilution of the sera used in ELISA? and what was the fold-dilution in the serial dilutions?
--> Responses
The starting dilution of immunized serum in the ELISA was 1:10,000. Serial dilutions were performed in two-fold dilutions starting at 1:10,000.
We have revised the legends for Fig. 1C, 1D, 2D, 2F, 3B, 4A, 5E, 5J, 6B, and 6C in the revised manuscript to accurately reflect this information.
The dilution factors of immunized sera or commercial antibodies for ELISA or Western blotting were described also in the revised figure legends.

Reviewer 2 Report
Comments and Suggestions for Authors
Viral infectious diseases, like COVID-19, can cause significant mortality, morbidity, and economic burdens for people. One of the most effective methods to prevent viral disease outbreaks is vaccines to prevent disease. This paper designed S2 peptide conjugated SARS-Cov-2 VLP vaccines. The study is very interesting and meaningful. However, the paper is recommended for publication after addressing the following minor comments:
1. What are the limitations of the newly designed S2 peptide-conjugated SARS-CoV-2 VLP vaccines?
2. What is the stability of the peptide-conjugated VLP vaccines?
3. How about the efficacy of this type of new vaccine? Does it need to add Adjuvant?
4. Table 1: Can the authors explain how the pI of the peptide and the hydrophilicity were calculated?
Comments on the Quality of English LanguageThe language in the results section is too tedious. Please shorten the section and make critical points about the results delivered to the readers.
Author Response
Question 1.
What are the limitations of the newly designed S2 peptide-conjugated SARS-CoV-2 VLP vaccines?
--> Responses
When inducing neutralizing antibodies against spike antigen, it is common to try to make full use of the structural antigen expressed by the virus. However, the production of recombinant viral structural proteins is not easy, and mRNA vaccines have been criticized for defective RNA reading frames, so it is important to propose methods to overcome these problems. Therefore, if epitopes that induce virus-neutralizing antibodies can be provided as linear peptides, it will contribute significantly to the production of safe vaccines, and our results are important in this regard.
As shown in the results, the content and level of antibody induction depends on how the important linear epitope is conjugated and displayed on the VLP. Therefore, it is necessary to adjust the epitope conjugation method and the length of the linker appropriately.
However, this study did not explore the diversity of epitope presentation methods. Additionally, the chemical conjugation method utilized is not well-suited for mass synthesis of VLP vaccines. Therefore, it is necessary to explore advanced methods that address these limitations. Currently, we are investigating the use of biological conjugation methods, specifically the SpyTag-SpyCatcher module, which facilitates conjugation through simple mixing.
Question 2.
What is the stability of the peptide-conjugated VLP vaccines?
--> Responses
Our study, described in this manuscript, was performed as part of a vaccine development study using VLPs, which focused on determining the antigenic peptide and conjugation method that can effectively induce antibodies that block antigenic sites on infectious SARS-CoV-2. Thus, we did not systematically validate the stability of the VLP vaccine for practical use in this manuscript.
However, after preparation of S2 peptide-conjugated VLPs, we confirmed that the VLP structure was maintained by TEM analysis, as shown in Fig. 2C, and that no aggregates were formed during storage of the peptide-conjugated VLPs at -80°C and thawing for injection.
To develop our results into practical applications, the physical stability, thermal stability, storage stability, and maintenance of immunogenicity of peptide-conjugated VLPs need to be further evaluated in terms of quality control. We are currently conducting research to develop effective VLP vaccines by diversifying the conjugation method of peptide antigens, the stability of peptide-conjugated VLPs will be considered in future studies.
Question 3.
How about the efficacy of this type of new vaccine? Does it need to add Adjuvant?
--> Responses
VLPs are far more immunogenic as compared to other subunit vaccines as they present repetitive antigenic epitopes on their surface in a more authentic confirmation that the immune system can readily detect. There is a report of 100% protection in adult mice against CHIKV infection when unadjuvanted CHIKV VLPs were used (PLoS Negl Trop Dis. 2019 Apr 26;13(4):e0007316).
In contrast, the polymer adjuvant induced a significantly higher HBsAg-specific IgG titer in mice than HBsAg-VLP alone after second immunization. The polymer adjuvant can effectively bound with HBsAg-VLP through electrostatic interactions to form a stable vaccine nanoformulation with a net positive surface charge. This nanoformulation had exhibited enhanced cellular uptake by macrophages. (ACS Appl Mater Interfaces. 2023 Oct 25;15(42):48871-48881).
While VLP antigen alone shows vaccine efficacy, it is important to consider that the local antigen storage effect and stability improvement provided by adjuvants also enhance vaccine efficacy.
In our study, we utilized the adjuvant TiterMax Gold (Sigma-Aldrich, Cat# T2684), which is a water-in-oil emulsion containing three essential ingredients (a block copolymer, CRL-8300, squalene, and a sorbitan monooleate) to enhance the immunogenic response.
VLP-based peptide vaccines, with their superior antigen presentation, are potentially effective as unadjuvanted vaccines; however, further studies are required.
Details regarding the use of this adjuvant are described in the "Materials and Methods" section under 2.5. Mouse Immunization of the revised manuscript (lines 265-267).
Question 4.
Table 1: Can the authors explain how the pI of the peptide and the hydrophilicity were calculated?
--> Responses
The isoelectric point (pI) and hydrophilicity of the peptides were calculated using tools provided on the website of Innovagen, a company specializing in peptide synthesis. The methodology used for this calculation is described on the Innovagen website (https://pepcalc.com/) and is cited in the section below Table 1 in the manuscript.
**Comments on the Quality of English Language
The language in the results section is too tedious. Please shorten the section and make critical points about the results delivered to the readers.
--> Responses
In the revised manuscript, we have made several changes as suggested by the reviewer. We have removed detailed descriptions that were not relevant to the experiments. We have also addressed the issues raised by the similarity check. To facilitate the review process, all deletions in the text have been struck through and corrections have been highlighted in red.
